



,

# Dependency of vertical velocity variance on meteorological conditions in the convective boundary layer

Noviana Dewani[1,2], Mirjana Sakradzija[2], Linda Schlemmer[3], Ronny Leinweber[4], and Juerg Schmidli[1,2]

[1]Institute for Atmospheric and Environmental Sciences, Goethe University Frankfurt, Germany
[2]Hans Ertel Centre for Weather Research, Deutscher Wetterdienst, Offenbach, Germany
[3]Deutscher Wetterdienst, Offenbach, Germany
[4]Deutscher Wetterdienst, Meteorologisches Observatorium Lindenberg – Richard-Aßmann-Observatorium, Lindenberg, Germany

**Correspondence:** Noviana Dewani (dewani@iau.uni-frankfurt.de)

**Abstract.** Measurements of vertical velocity from vertically pointing Doppler lidars are used to derive the profiles of vertical velocity variance. Observations were taken during the FESSTVaL (Field Experiment on Submesoscale Spatio-Temporal Variability in Lindenberg) campaign during the warm seasons of 2020 and 2021. Normalized by the square of convective velocity scale, the average vertical velocity variance profile follows the universal profile of Lenschow et al. (1980), however, daily profiles still show a high day-to-day variability. We found that moisture transport and the content of moisture in the boundary layer could explain the remaining variability of the normalized vertical velocity variance. The magnitude of the normalized vertical velocity variance is highest on clear-sky days, and decreases as the relative humidity increase and surface latent heat flux decrease in cloud-topped and rainy days. This suggests that moisture content and moisture transport are limiting factors for the intensity of turbulence in the convective boundary layer. We also found that the intensity of turbulence decreases with an increase in boundary layer cloud fraction during FESSTVaL, while the latent heating in the cloud layer was not a relevant source of turbulence in this case. We conclude that a new vertical velocity scale has to be defined that would take into account the moist processes in the convective boundary layer.

## 1 Introduction

Turbulence has an important role in distributing heat, momentum, moisture and trace gases from the land surface to the free troposphere. As a measure of the intensity of turbulent structures in a convective boundary layer, such as updrafts or thermals, vertical velocity variance, $\sigma_w^2$, is frequently used. Vertical velocity variance normalized by the square of the convective velocity scale (Deardorff, 1970), $\sigma_w^2/w_*^2$, has been studied using both observational data (e.g. Hogan et al., 2009; Maurer et al., 2016) and numerical models (e.g. Lenschow et al., 2012; Zhou et al., 2019). The previous studies consistently show that the mean vertical profile of $\sigma_w^2/w_*^2$ follows the universal function introduced by Lenschow et al. (1980):

$$\sigma_w^2/w_*^2 = 1.8(z/z_i)^{2/3}(1 - 0.8(z/z_i))^2 \tag{1}$$





where $z$ represents height and $z_i$ is the depth of the mixed layer. This function gives an asymmetric vertical profile with a maximum of the $\sigma_w^2/w_*^2$ at about $0.3z_i$ - $0.4z_i$. The universal profile was derived from in situ aircraft measurement data during AMTEX (Air Mass Transformation Experiment) that took place over the East China Sea during wintertime cold air outbreaks.

Although there is universality in the mean $\sigma_w^2/w_*^2$ profile across many case studies and seasons in both clear-sky and cloud-topped boundary layers, the variability of daily profiles of $\sigma_w^2/w_*^2$ is high and their dependency on the boundary layer conditions varies from case to case. A considerable scatter of the $\sigma_w^2/w_*^2$ profiles was found in the study of Hogan et al. (2009), which analysed profiles of two clear-sky days and four shallow-cumulus days. They found that the mean profile was similar to the universal profile of Lenschow et al. (1980) with no significant difference between the profiles in clear-sky and cloud-topped
days. Lareau et al. (2018) conducted a study at the ARM Southern Great Plains (SGP) site in Oklahoma, United States, and found significantly different behaviour of $\sigma_w^2/w_*^2$ compared to the previous study of Chandra et al. (2010) conducted at the same location. A higher magnitude of $\sigma_w^2/w_*^2$ was found on days with a higher cloud fraction in Chandra et al. (2010), while Lareau et al. (2018) found the highest magnitude of $\sigma_w^2/w_*^2$ at an intermediate range of cloud fraction. Moreover, Lareau et al. (2018) observed a lower magnitude of $\sigma_w^2/w_*^2$ on clear-sky days compared to the cloud-topped days, opposite to Chandra et al.
(2010) where the largest magnitude of $\sigma_w^2/w_*^2$ was found in the clear-sky category. In a year-long data set from the same site (ARM SGP), Berg et al. (2017) found a sensitivity of the $\sigma_w^2/w_*^2$ magnitude on clear-sky days to the season, friction velocity, stability and wind shear across the boundary-layer top. An earlier study of Lenschow et al. (2012) also found a considerable residual scatter after normalization in the daily profiles of the $\sigma_w^2/w_*^2$, with about 10% of the variations that could not be explained by the effects of wind shear, stability, or the variability in land surface properties. Furthermore, during the days with
mesoscale circulations, such as longitudinal roll circulations, the peak of the $\sigma_w^2/w_*^2$ profile was lifted to about $0.6\,z_i$ - $0.7\,z_i$ even when the surface heat flux values remained comparable to the other cases (Lenschow et al., 2012).

The following research questions stem from the previous studies: Where does the residual variation in the daily profiles after normalization come from? Is cloud fraction a relevant parameter to study the changes in the magnitude of $\sigma_w^2/w_*^2$ from case to case? Are boundary layer clouds a significant source of turbulence in the convective boundary layer?

Various observational methods can be used to obtain the variance of vertical velocity measurements. In the cited studies, most of the vertical velocity data were obtained by in situ aircraft measurement but the measurements were limited to the height and the number of flights. The conventional meteorological tower using sonic anemometer can be used to obtain vertical velocity variance (e.g. Bonin et al., 2016). However, in this case, the height of the retrieval is also limited depending on the height of the tower. These limitations of the earlier measurement techniques are overcome by the advantages of ground-based
remote sensing using Doppler lidars. Doppler lidars are able to measure continuously and cover the entire boundary layer depth. Besides vertical velocity measurement, Doppler lidars have been used to measure wind speed and wind direction (e.g. Päschke et al., 2015), wind gusts (e.g. Suomi et al., 2017), turbulence (e.g. Sathe et al., 2015; Smalikho and Banakh, 2017), and identify coherent structures (e.g. Ansmann et al., 2010; Cheliotis et al., 2020). Doppler lidars are a reliable method to retrieve the $\sigma_w^2/w_*^2$ profile, as shown in a comparison between $\sigma_w^2/w_*^2$ profile derived from a Doppler lidar, Large Eddy Simulations
(LES) and the empirical profile (Lenschow et al., 2012).





In this study, we investigate the dependency of $\sigma_w^2/w_*^2$ on the meteorological parameters using Doppler lidar measurements. The aim is to find the key parameters that explain the variability of $\sigma_w^2/w_*^2$ that could be used in the future to derive a scaling velocity taking into account the missing factors controlling the intensity of turbulence in the convective boundary layer. The Doppler lidar data was collected during two consecutive summer periods of the FESSTVaL (Field Experiment on Submesoscale Spatio-Temporal Variability in Lindenberg) campaign (https://fesstval.de/), from June-August 2020 and May-August 2021. The measurement campaigns aimed at identifying sub-mesoscale variability, such as atmospheric boundary layer structure, cold pools, and wind gusts, and took place at the Meteorological Observatory Lindenberg - Richard-Aßmann-Observatorium (MOL-RAO) of the German Weather Service (DWD) near Berlin. The structure of this paper is as follows: we describe the instruments and the measurements in section 2. The selected days and case categories are described in section 3. In sections 4 and 5, we present the results and a discussion, followed by the conclusion at the end of the paper.

## 2    Data, method and measurements

Two different units of the Halo Photonics Streamline XR Doppler LiDAR set up in a vertical stare configuration were used to measure the vertical velocity during FESSTVaL. In the 2020 measurement campaign, the Halo Photonics Streamline XR 161 (DL161) was used while in the 2021 measurement campaign, we used Halo Photonics Streamline XR 146 (DL146). The details of specifications are shown in Table 1. Besides the Doppler lidar data, the routine measurements from the Falkenberg site are used in this study. The surface heat flux for the calculation of the convective velocity scale were obtained from the eddy-covariance measurements using a sonic anemometer (USA-1, METEK GmbH) and an infrared gas analyzer (LI7500RS, Licor Inc.) at a height of 2.4 m. We also used friction velocity ($u_*$) retrieved from the same instrument for the analysis in Section 4. The 10 m wind speed, relative humidity and temperature were obtained from the 99 m meteorological mast which is located near the Doppler lidar position as shown in Fig. 1 using Thies cup anemometer and Vaisala HMP-45. In addition, precipitation measurements were used to sort out the rainy days in the collected datasets and the ceilometer data from CHM15k-080066 instrument to make a comparison to the cloud detection from Doppler lidar data.

For Doppler lidar data quality control, different signal-to-noise-ratio (SNR) filters were applied to the two datasets. For DL161 data, we applied 1.005 as a threshold of the backscatter intensity (SNR+1) parameter in the first filtering step. The following additional procedure has been applied to the DL161 data after the SNR filtering. The tested data point is removed if the difference between the tested data to the surrounding data in an 8x8 matrix is more than $5\,\mathrm{m\,s^{-1}}$. While for the DL146 data, a different SNR+1 threshold within a range between 0.994 and 1.005 was applied on each day in order to obtain more data at the highest height level. The threshold is determined as a limit close to the data which is no longer distributed over the search band ($\pm\,19\,\mathrm{m\,s^{-1}}$) as shown in Fig. 2. Besides the SNR filter, the first two elevation levels in DL146 and first four elevation levels in DL161 datasets were removed due to the high noise level in these lowest range gates. Therefore, the lowest levels are 81 m in the DL161 and 120 m in the DL146. To ensure comparability between the two different Doppler lidar units, an intercomparison between DL161 and DL146 was performed on 23-25 July 2021 in vertical stare mode. The measurements of the two different Doppler lidars showed a high correlation and thus a good agreement at 120 m height (Fig. 3).





**Table 1.** Technical specification of Doppler lidars DL161 and DL146

| Halo Streamline | DL161 | DL146 |
|---|---|---|
| Wavelength | $1.5\,\mu m$ | $1.5\,\mu m$ |
| Pulse Range Frequency (PRF) | $10\,\mathrm{kHz}$ | $10\,\mathrm{kHz}$ |
| Vertical resolution | $18\,\mathrm{m}$ | $48\,\mathrm{m}$ |
| Minimum range gate | $9\,\mathrm{m}$ | $24\,\mathrm{m}$ |
| Maximum range gate | $3\,\mathrm{km}$ | $5\,\mathrm{km}$ ($12\,\mathrm{km}$*) |
| Lowest usable range gate | $81\,\mathrm{m}$ | $120\,\mathrm{m}$ |
| Pulse width | $330\,\mathrm{ns}$ | $413\,\mathrm{ns}$ |
| Time resolution | $1.5\,\mathrm{s}$ | $3\,\mathrm{s}$ |

*period after 12 August 2021

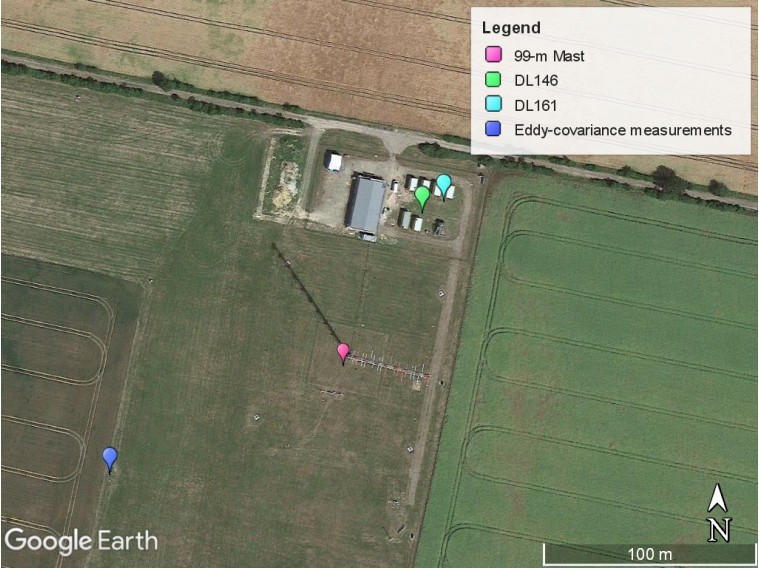

**Figure 1.** Measurement location in Falkenberg, MOL-RAO (source: © Google Earth).

The vertical velocity variance was calculated in $30\,\mathrm{min}$ averages using the method from Lenschow et al. (2000) to remove
uncorrelated noise in the higher-order statistics. First, auto-covariance is calculated for the first 40 points. Next, the corrected
variance was estimated by extrapolating the auto-covariance at zero-lag by linear extrapolation, excluding the lag zero. The
difference between the variance at zero-lag and the uncorrected variance is the uncorrelated noise.

For the analysis, the vertical velocity variance was normalized by the convective velocity defined by the equation

$$w_* = [gz_i(SHF + 0.7LHF)/(T_v\rho c_p)]^{1/3} \tag{2}$$





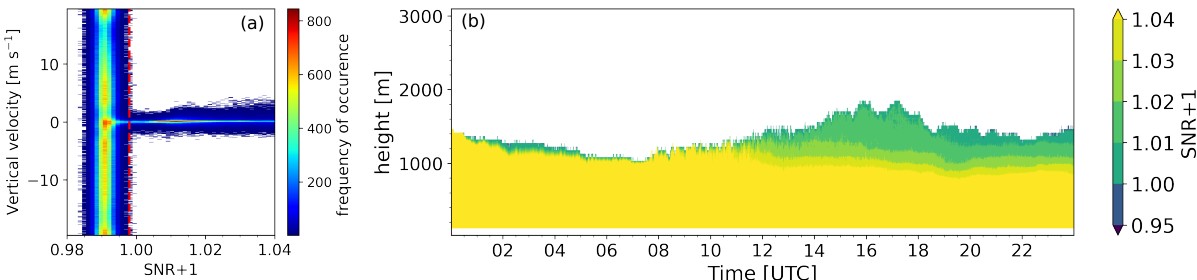

**Figure 2.** (a) 2D histogram of intensity (SNR+1) and vertical velocity on 14 June 2021. The red line indicates the threshold used on 14 June 2021 data; (b) the intensity plot using the determined threshold.

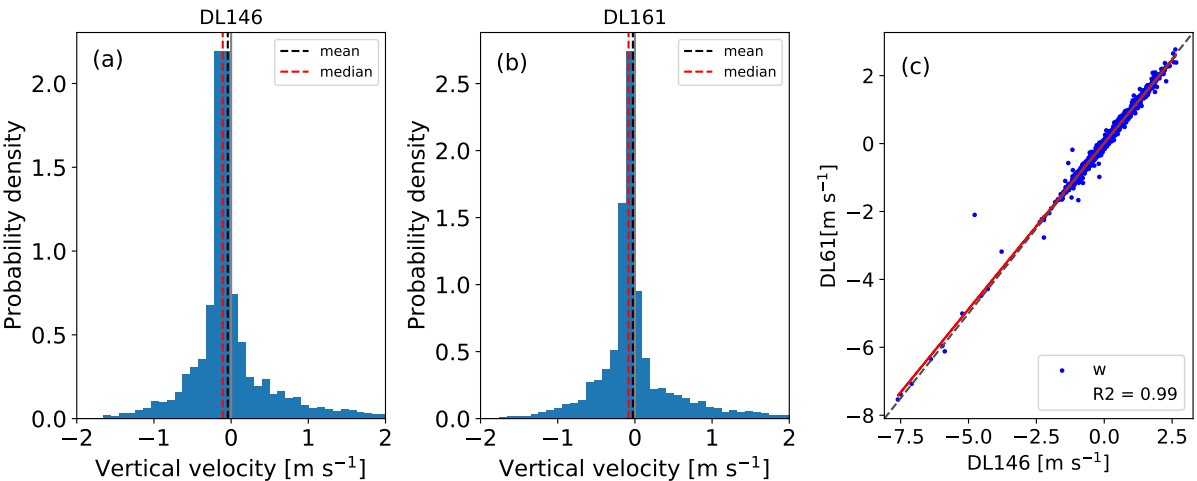

**Figure 3.** Histograms of vertical velocity of (a) DL146 and (b) DL161 at 120 m height based on measurements taken between 23-25 July 2021. (c) scatterplot of vertical velocity for DL161 and DL146.

where $g$ is gravity, $z_i$ is the estimated mixing layer height, $SHF$ and $LHF$ are the surface sensible and latent heat fluxes, $c_p$ is the specific heat capacity, $T_v$ is the virtual temperature, and $\rho$ is the air density. Besides the normalization of vertical velocity variance, the height also has to be normalized by the boundary layer height. In both datasets, the mixing layer height was estimated using the variance method (Tucker et al. (2009)) with different thresholds, $0.04\,\mathrm{m^2\,s^{-2}}$ for the DL161 dataset and $0.09\,\mathrm{m^2\,s^{-2}}$ for the DL146 dataset. The mixed layer top is estimated as the first layer at which the variance is below the

threshold. An example of the estimated mixing layer height is shown in Fig. 4 by the black dashed line. The clouds identified based on the Doppler lidar data (green) and ceilometer data (cyan) show good agreement.





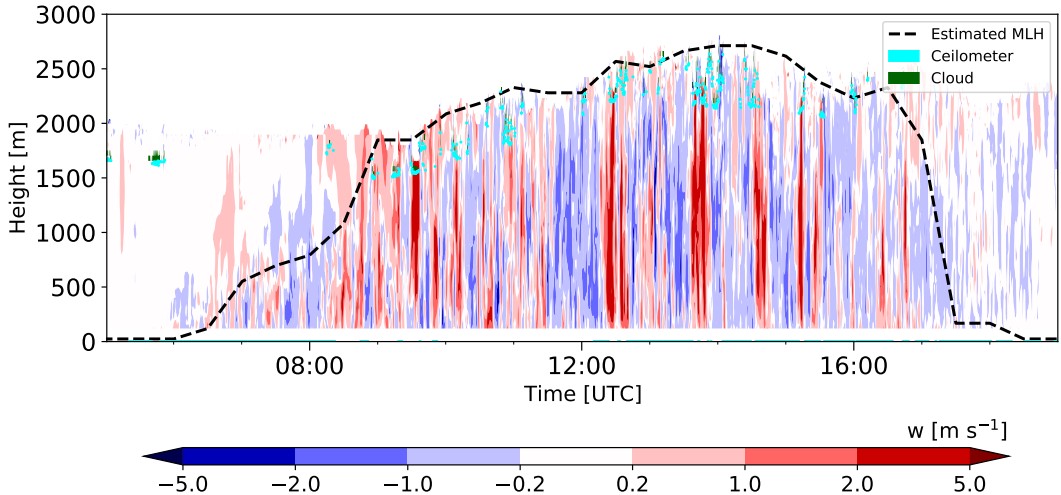

**Figure 4.** Vertical velocity averaged over 1 min on 11 June 2021. The black dashed line indicates the estimated mixing layer height based on the Doppler lidar measurements, the green dot shows the clouds from attenuated backscatter coefficient from the Doppler lidar, and the cyan dot indicates the cloud base height based on ceilometer measurements.

## 3  Data selection and categorization

Three main categories were created based on the presence of clouds: clear-sky, cloud-topped and rainy days. Clouds are detected using attenuated-backscatter from Doppler lidar data employing a threshold of $10^{-4}$ m$^{-1}$ $sr^{-1}$. The hourly cloud fraction is used to categorize clear-sky and cloud-topped days. Days with a cloud fraction above 0.003 were categorized as cloud-topped. This study considers the low clouds but does not consider the type of cloud and focuses more on finding a general dependency of the $\sigma_w^2/w_*^2$ profile on cloud fraction. The category of rainy days is defined when the precipitation is recorded during the day.

The analysis used a total of 88 selected days from two consecutive summer periods, consisting of 11 clear-sky days, 59 cloud-topped days, and 18 rainy days. Specifically for rainy days, the rain period was excluded from the analysis. An example of 1 min average vertical velocity and 30 min average of the vertical velocity variance for each category is shown in Fig. 5.





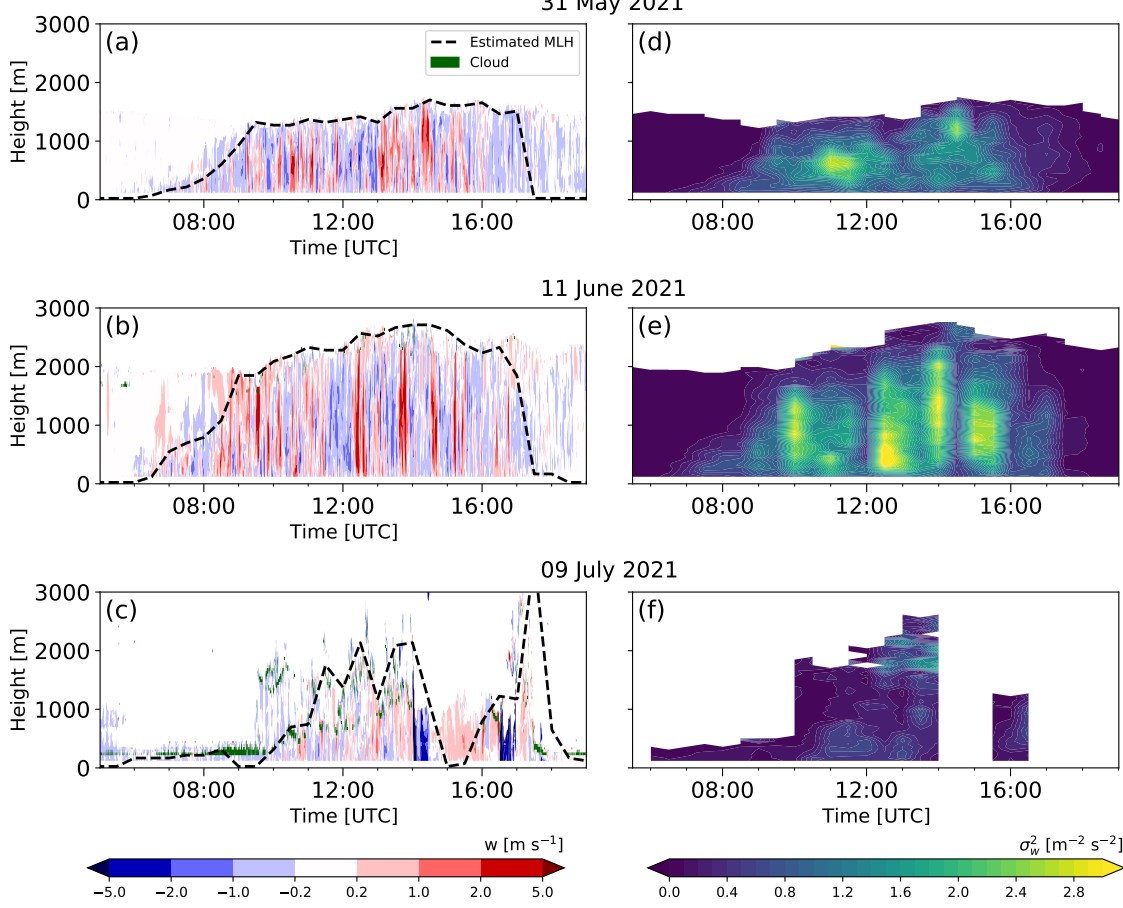

**Figure 5.** Example of (a), (b), (c) one-minute averages of vertical velocity and (d), (e), (f) 30 min averages of variance of vertical velocity on a clear-sky day (31 May 2021), cloud-topped day (11 June 2021), and rainy day (09 July 2021). The estimated mixing layer height based on the Doppler lidar measurements is indicated by a black dashed line, while the cloud layer is given in green.

The analysis is based on the time period from 10 UTC to 15 UTC with the assumption that the boundary layer is in a relatively well developed stage during this period. Figure 6 shows the evolution of the mixing layer height on clear-sky days and cloud-topped days from the 30 min composites in each case. A higher boundary layer height is observed on cloud-topped days compared to clear-sky days. The composite of hourly $\sigma_w^2/w_*^2$ profiles from combined clear-sky and cloud-topped days is shown in Fig. 7a. The magnitude of $\sigma_w^2/w_*^2$ profiles increases from the morning hours as the boundary layer starts to develop to about 10 UTC. During the day when the boundary layer was well developed, between 10 UTC and 15 UTC, the magnitude of the $\sigma_w^2/w_*^2$ profile becomes less variable. After 15 UTC, the magnitude of the $\sigma_w^2/w_*^2$ profile increases again as the surface heat flux decreases and the boundary layer starts to collapse in the afternoon hours.





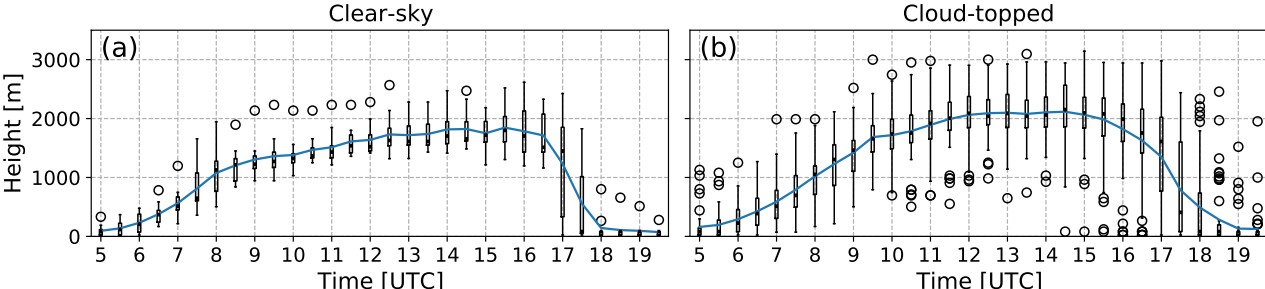

**Figure 6.** Composite of 30 min mixing layer height for (a) clear-sky days and (b) cloud-topped days. The blue line indicates the mean of mixing layer height while the box denotes upper quartile and lower quartile with the whiskers shows the extension of 1.5x of the inter-quartile range. The outliers is denoted by the circles.

## 4 Results

### 4.1 Vertical velocity variance

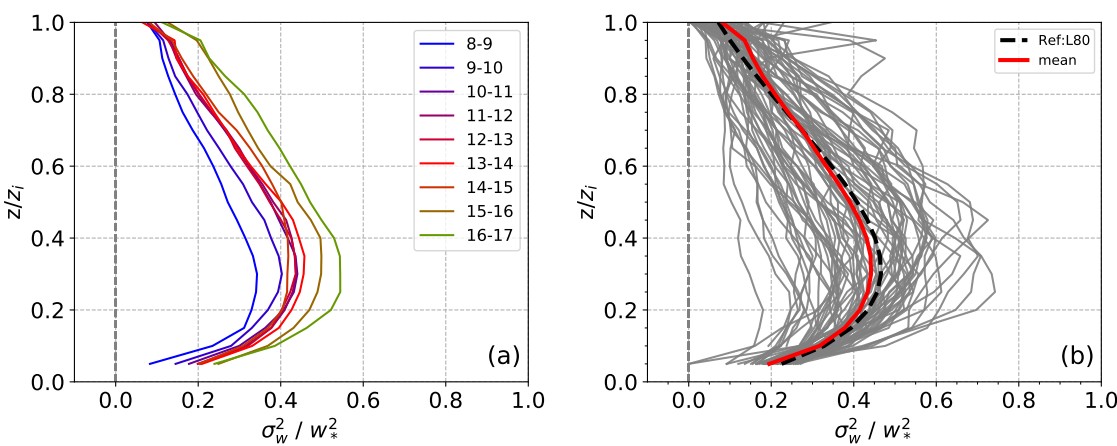

**Figure 7.** (a) Diurnal change of the composite hourly profiles of $\sigma_w^2/w_*^2$ from clear-sky and cloud-topped days combined; (b) daily profiles of $\sigma_w^2/w_*^2$ showing individual days (grey line) and the mean profile (red line). The universal profile of Lenschow et al. (1980) is given as black dashed line.

The mean $\sigma_w^2/w_*^2$ profile from the selected clear-sky and cloud-topped days is similar to the empirical universal profile as shown in Fig. 7b. However, as in the previous studies, there is considerable variation in the day-to-day $\sigma_w^2/w_*^2$ profiles. This variability remains after scaling the vertical velocity variance with $w_*^2$, which signifies that there is another relevant factor that controls the intensity of turbulence in the convective boundary layer, other than the generation of turbulence by buoyancy, that





is not accounted for. Furthermore, we also tested the effect of shear included in the scaling velocity introduced by Moeng and Sullivan (1994) and found no significant reduction in the variability of the day-to-day $\sigma_w^2/w_*^2$ profiles (not shown).

The difference in mean $\sigma_w^2/w_*^2$ profiles of the three main categories, clear-sky, cloud-topped and rainy days, is shown in Fig. 8. The magnitude of $\sigma_w^2/w_*^2$ is highest during the clear-sky days, while it decreases on cloud-topped days and has the lowest

value during the rain-free periods of the rainy days. The average profiles of the three categories show a clear dependency on the relative humidity in the boundary layer, as the $\sigma_w^2/w_*^2$ decreases with an increase in relative humidity (Fig. 8d). The surface latent heat flux is decreasing while relative humidity near the surface is increasing from the clear-sky to the rainy days category (Fig. 8b and Fig. 8c). The mean surface latent heat flux does not significantly differ between clear-sky and cloud-topped days. Nevertheless, the range of surface latent heat flux is larger on cloud-topped days including very low surface latent

heat flux values. The same behavior was seen in the range of relative humidity values at 10 m height which has a larger range on the cloud-topped days than on the clear-sky days. To investigate where the variability originates from on clear-sky and cloud-topped days, we look at the influence of different meteorological parameters on the $\sigma_w^2/w_*^2$ profiles in the following.

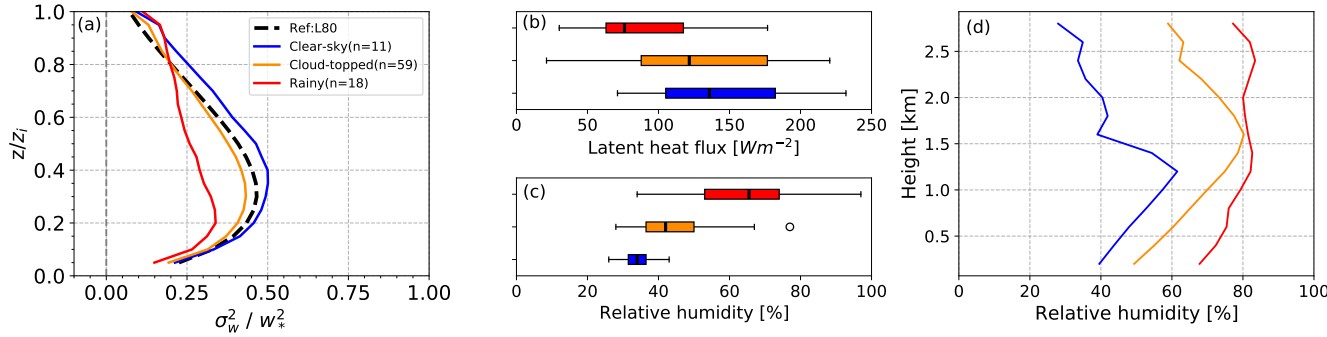

**Figure 8.** (a) Average profiles of $\sigma_w^2/w_*^2$ of the three main categories: clear-sky, cloud-topped and rainy days. The universal profile is given as black dashed line. Box plot of the (b) surface latent heat flux (LHF) and (c) relative humidity (RH) averaged by the three categories; (d) vertical profile of relative humidity at 12 UTC in the boundary layer averaged over the three main categories from radiosonde data (https://weather.uwyo.edu/upperair/sounding.html).





## 4.2 Factors that control the vertical velocity variance

### 4.2.1 Clear-sky boundary layer

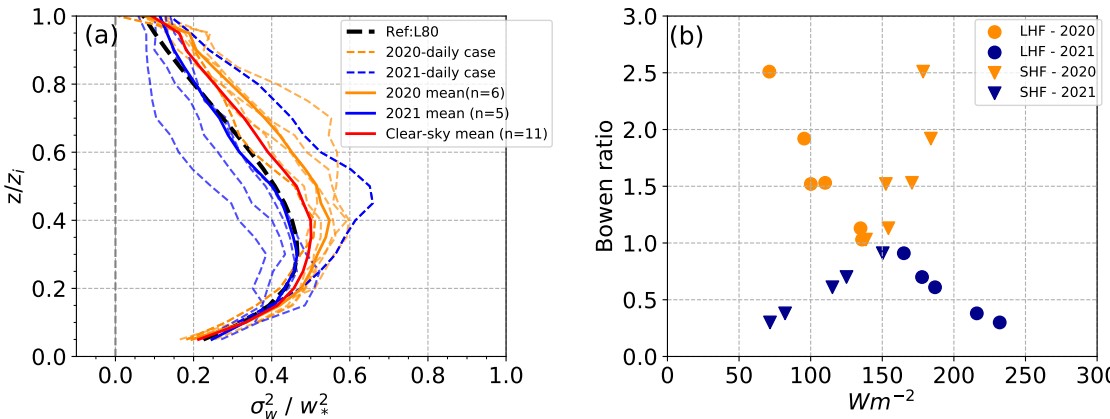

**Figure 9.** (a) $\sigma_w^2/w_*^2$ profiles averaged from 10 UTC-15 UTC by individual days in the clear-sky category; (b) surface sensible heat flux (triangle) and surface latent heat flux (circle) for the 2020 (orange) and the 2021 (blue) datasets.

The variability of daily $\sigma_w^2/w_*^2$ over the 11 clear-sky days shows a clear distinction between the datasets of the two years, 2020 and 2021 (Fig. 9a). We found that this distinction results from a significant difference in the surface Bowen ratio (BR) between the 2020 and 2021 datasets as shown in Fig. 9b. In the 2020 dataset, the Bowen ratio value is larger (BR higher than about 1) compared to the 2021 dataset (BR lower than 1), indicating drier surface conditions in 2020. In the data set with the lower Bowen ratio, the magnitude of $\sigma_w^2/w_*^2$ is lower, and vice versa. One exception is a single outlier case on 31 May 2021 which

was most likely driven by synoptic-scale patterns. Besides the Bowen ratio, other meteorological parameters do not show any systematic pattern for $\sigma_w^2/w_*^2$, such as the bulk stability and friction velocity. Within these clear-sky days, the bulk stability parameter falls in range between $-z_i/L = 52$ and $-z_i/L = 60$ and the friction velocity has a range of 0.21-0.38 m s$^{-1}$.





#### 4.2.2 Cloud-topped boundary layer

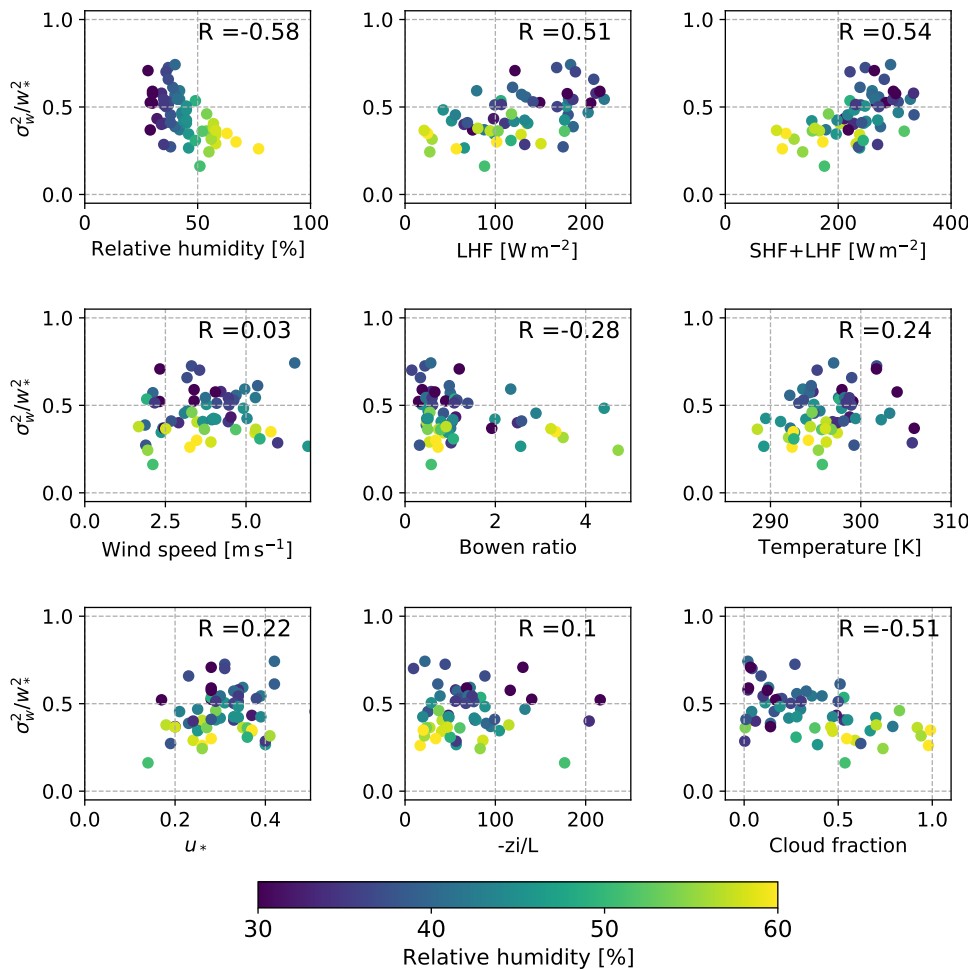

**Figure 10.** Correlation between several meteorological parameters and averaged values of $\sigma_w^2/w_*^2$ between $0.25\,z_i$ and $0.6\,z_i$. The color of dots indicates the value of relative humidity. A high correlation is shown in the relative humidity, total surface heat flux, surface latent heat flux (LHF), cloud fraction and surface Bowen ratio.

The dependency of $\sigma_w^2/w_*^2$ averaged in the layer between $0.25\,z_i$ and $0.6\,z_i$ on several meteorological parameters in a cloud-topped boundary layer is examined in Fig. 10. The averaged $\sigma_w^2/w_*^2$ shows an absolute correlation coefficient equal to 0.3 with Bowen ratio and higher than 0.5 with cloud fraction, relative humidity, surface latent heat flux and total surface heat flux. In addition, the dots are colored by their relative humidity to demonstrate the joint dependencies. The parameters that show a high correlation also show a clustering of points with similar relative humidity. A cluster with high relative humidity is found at relatively high cloud fraction values and also at lower total heat flux. Based on these significant parameters, we further classify the cloud-topped days by cloud fraction, surface latent heat flux, relative humidity and Bowen ratio.





Figure 11a shows the profiles of $\sigma_w^2/w_*^2$ averaged over the days that fall into three ranges of cloud fraction, using 0.25 and 0.5 as threshold values. As the cloud fraction increases, the $\sigma_w^2/w_*^2$ magnitude becomes smaller. As expected, the cloud fraction categories show a systematic change in relative humidity. High relative humidity in high cloud fraction is associated with a low magnitude of the $\sigma_w^2/w_*^2$, similarly to the three main categories in Fig. 8. The Bowen ratio does not significantly differ

between the three cloud fraction ranges. In the case of high cloud fraction, the surface latent heat flux value is lower, while the two other categories do not show such a clear distinction in the surface latent heat flux.

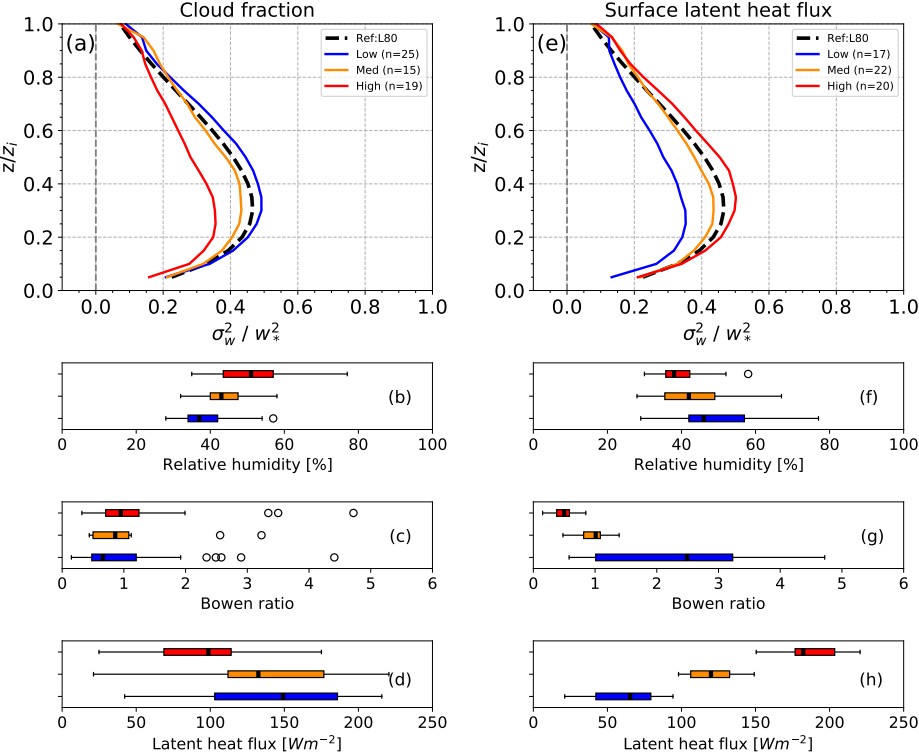

**Figure 11.** Vertical profiles of $\sigma_w^2/w_*^2$ averaged based on different classifications: (a) cloud Fraction; (e) surface latent heat flux. In the box plots, the corresponding values of relative humidity ((b) and (f)); Bowen ratio ((c) and (g)) and surface latent heat flux ((d) and (h)) for each classification are shown. The number of days (n) in each category are added in the legend.

As one of the driving factors of the turbulence, the dependency of $\sigma_w^2/w_*^2$ on the surface heat flux is investigated. Since the sensible heat flux is already taken into account in the convective velocity scale, the $\sigma_w^2/w_*^2$ dependency is now examined based on the surface latent heat flux classification. The collected data is divided into three categories, low (LHF $< 98\,\mathrm{W\,m^{-2}}$), 

medium ($98\,\mathrm{W\,m^{-2}} \leq$ LHF $< 150\,\mathrm{W\,m^{-2}}$ and high (LHF $\geq 150\,\mathrm{W\,m^{-2}}$) latent heat flux. Figure 11e shows that the $\sigma_w^2/w_*^2$ profiles systematically increase from low to high surface latent heat flux. This is consistent with the result of the cloud fraction classification where lower $\sigma_w^2/w_*^2$ is also associated with lower surface latent heat flux values. Relative humidity is the lowest for the highest surface latent heat flux category and vice versa. This is also consistent with the result shown in Fig. 8 where the





highest magnitude of $\sigma_w^2/w_*^2$ if found for the lowest relative humidity values. It is interesting to observe that the Bowen ratio
has values higher than 1-1.5 only if the surface latent heat flux is lower than the selected threshold of $98\,\mathrm{W\,m^{-2}}$.

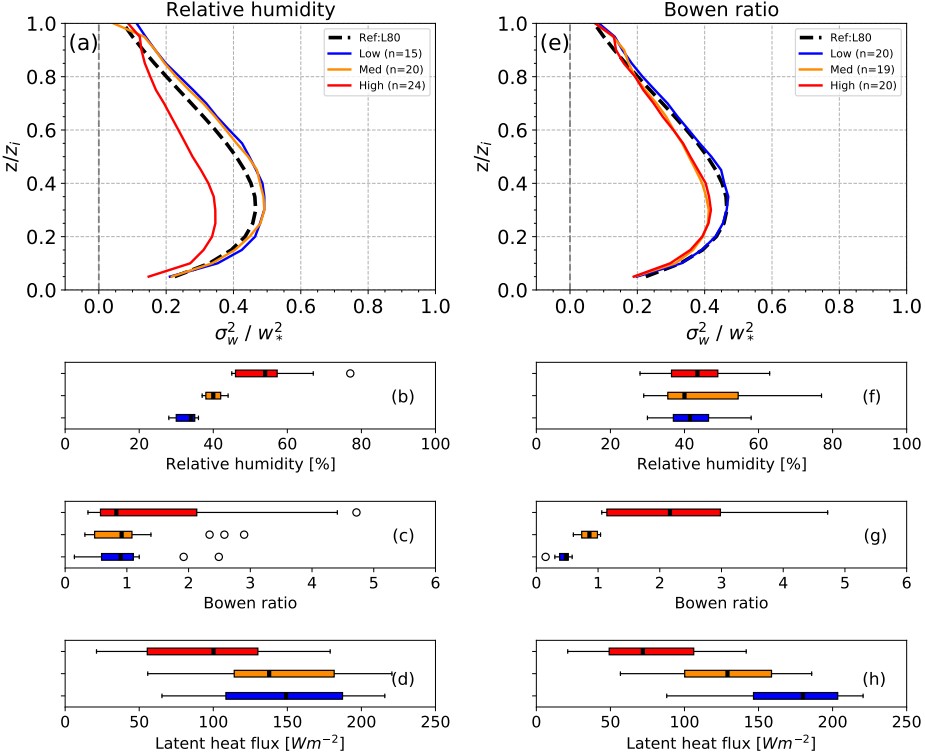

**Figure 12.** Similar as in Fig. 11, but for (a) relative humidity; (e) Bowen ratio.

Figure 12a shows the $\sigma_w^2/w_*^2$ profiles averaged over days that fall into the three ranges of relative humidity, low (RH <
37 %), medium ($37\,\% \leq$ RH $< 45\,\%$) and high (RH $\geq 45\,\%$). The $\sigma_w^2/w_*^2$ shows a larger magnitude in the range of relative
humidity below 45 %, while a lower magnitude of $\sigma_w^2/w_*^2$ is found in the high relative humidity ranges. This pattern can also
be seen in the clear-sky, cloud-topped and rainy days comparison in Fig. 8 and also in the cloud fraction classification Fig. 11a.
The dependency of the averaged $\sigma_w^2/w_*^2$ on the Bowen ratio is not as clear as on clear-sky days. The $\sigma_w^2/w_*^2$ profiles averaged
over the days with high and medium Bowen ratio are similar, while the days with a lower Bowen ratio have a higher magnitude
of $\sigma_w^2/w_*^2$ (Fig. 12e). In the cloud-topped boundary layer, the dependency of the profiles on the Bowen ratio is opposite to the
dependency found on clear-sky days. This could be the result of a significantly higher relative humidity on cloud-topped days,
as the magnitude of $\sigma_w^2/w_*^2$ is lower in the case of higher relative humidity.
We did not find any considerable dependency of $\sigma_w^2/w_*^2$ on other parameters used to characterize turbulence, such as bulk
stability. We used the bulk stability, $-z_i/L$, where $L$ is the Obukhov length. The same threshold as in Lenschow et al. (2012) is
applied with $-z_i/L < 30$ for the less unstable and $-z_i/L \geq 30$ for the more unstable category. From the total of 58 days used
for this case, 10 days are in the less unstable category and 48 days are in the more unstable category. Even though the range of





the values of the stability parameter is similar to the range in Lenschow et al. (2012), we found similar profiles of $\sigma_w^2/w_*^2$ in

the two categories (Fig. 13). On the other hand, there is a dependency of $\sigma_w^2/w_*^2$ on the friction velocity ($u_*$), however it does not show a systematic pattern. The $\sigma_w^2/w_*^2$ profiles are divided into three categories based on the friction velocity: low ($u_* < 0.27\,\mathrm{m\,s^{-1}}$), medium ($0.27\,\mathrm{m\,s^{-1}} \leq u_* < 0.34\,\mathrm{m\,s^{-1}}$, high ($u_* \geq 0.34\,\mathrm{m\,s^{-1}}$). The lowest magnitude of $\sigma_w^2/w_*^2$ is found for the low friction velocity, while the highest magnitude of $\sigma_w^2/w_*^2$ is found for the medium friction velocity.

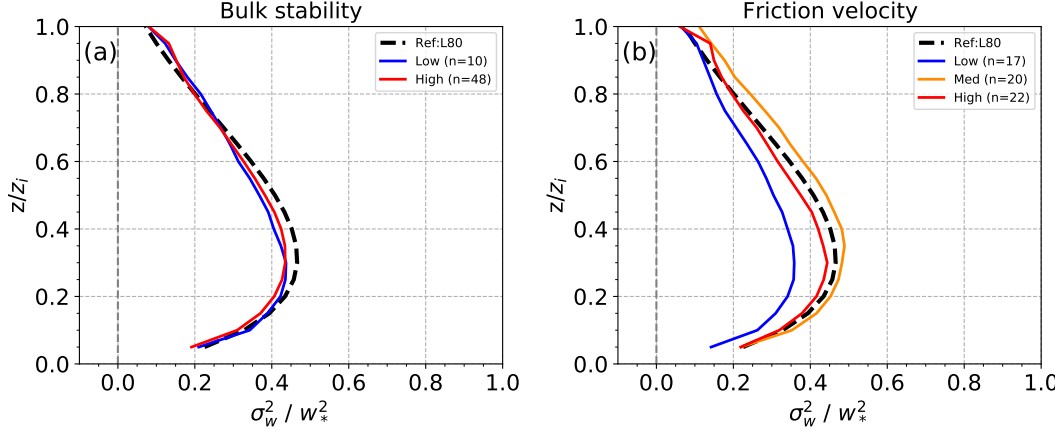

**Figure 13.** Classification of $\sigma_w^2/w_*^2$ based on the (a) bulk stability and (b) friction velocity. In the legend, $n$ indicates the number of days in each class.

### 4.2.3 Rainy days

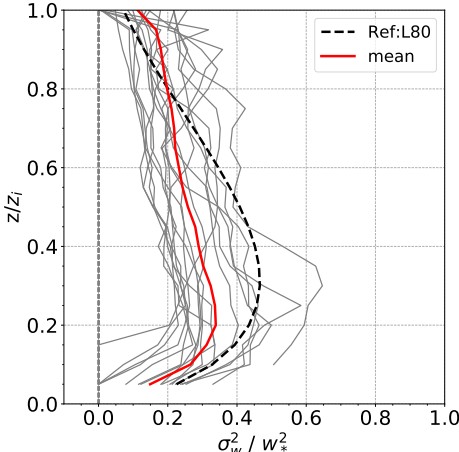

**Figure 14.** Vertical profiles of daily $\sigma_w^2/w_*^2$ averaged from 10 UTC to 15 UTC excluding the precipitation periods collected from 18 rainy days. The red line indicates the mean profile and the black dashed line indicates the empirical profile (Lenschow et al. (1980)).





Daily average profiles of $\sigma_w^2/w_*^2$ during the periods before or after rain are shown in Fig. 14. The mean $\sigma_w^2/w_*^2$ profile has a smaller magnitude compared to clear-sky and cloud-topped days with a maximum value of $\sigma_w^2/w_*^2$ at $0.2\,z_i$. The lower height of maximum values might be due to the weaker updraft and downdraft in this case. Moreover, the day-to-day variability of the daily profiles of $\sigma_w^2/w_*^2$ is lower compared to the other categories.

## 5   Discussion

Daily averaged profiles of $\sigma_w^2/w_*^2$ on clear-sky days are strongly related to the values of the surface Bowen ratio. Since the profiles are normalized by the convective velocity scale ($w_*$), the dependency of $\sigma_w^2/w_*^2$ on the surface Bowen ratio is introduced through the changes in the surface latent heat flux and cannot be explained by the generation of turbulence by buoyancy. The $\sigma_w^2/w_*^2$ was lower during days with a higher surface latent heat flux and vice versa. This suggests that a higher flux of moisture from the land surface might place a limit to the strength of convective circulations that maybe act as a

dehumidifier of boundary layer in these cases similar to the case of deep convection (Pauluis and Held, 2002), however, this is only a speculation and needs further investigation. One outlier in the clear-sky 2021 dataset, 31 May 2021, shows a larger magnitude of $\sigma_w^2/w_*^2$ with maximum values at $0.5\,z_i$. Compared to the other sample days in 2021, the higher soil moisture content, around $17\%$, is found on that day, while all other days had soil moisture content well below $14\%$. We also observed a higher magnitude of the mean $\sigma_w^2/w_*^2$ on clear-sky days compared to the result in study of Berg et al. (2017) which focus on

the clear-sky days in a year-long dataset and also the result in Lareau et al. (2018) which included clear-sky days in their study.

    We looked into the dependency of $\sigma_w^2/w_*^2$ on cloud fraction, as the previous studies showed contrasting results. We found that the magnitude of $\sigma_w^2/w_*^2$ decreased with an increase in cloud fraction, which is the opposite to the results of the previous studies. While Hogan et al. (2009) found no significant difference between $\sigma_w^2/w_*^2$ on clear-sky and cloud-topped days, Chandra et al. (2010) found higher $\sigma_w^2/w_*^2$ on days with a higher cloud fraction. On the other hand, Lareau et al. (2018) found the highest

$\sigma_w^2/w_*^2$ at an intermediate cloud fraction. Our results thus confirm the conclusion made in Hogan et al. (2009) that boundary-layer clouds are not a relevant source of the turbulence in the convective boundary layer, moreover, our results suggest the opposite, that formation of clouds acts as a sink rather than a source of turbulence. We also show a similar dependency in the $\sigma_w^2/w_*^2$ profile in the comparison between clear-sky, cloud-topped and rainy days where the $\sigma_w^2/w_*^2$ was decreased from clear-sky to rainy days.

The results of our study related to cloud-topped days should not be generalized to other locations or seasons. The development of boundary layer clouds involves a number of complex and competing mechanisms of the land-atmosphere system. These interactions manifest in a non-linear dependency of cloud development on the state of soil moisture. So, for example, if the atmospheric stability above the boundary layer is strong, soil moisture acts to support the cloud development. However, if the atmospheric stability is weak, clouds will be favoured over dry soils, while the increase of soil moisture will act to decrease

the probability of cloud development (Ek and Holtslag, 2004; Gentine et al., 2013). The intensity of convective updrafts that form the clouds is thus more likely related to the state of soil moisture and the magnitude of the surface heat flux and not the cloud fraction itself. This intricate dependency of the formation of clouds on soil moisture might explain why we find





such a difference between our results at the Lindenberg observatory and the previous result from the ARM GPS site, the two observational sites that have a very distinct regime of soil moisture: the first one is generally abundant in soil moisture, while the second one is in a semi-arid regime (Koster et al., 2004).

The contrasting results in the dependency of $\sigma_w^2/w_*^2$ on the surface Bowen ratio in clear-sky and cloud-topped days in our study (Fig. 9 and Fig. 12e) can be explained by the compounding dependency of $\sigma_w^2/w_*^2$ on relative humidity. During the clear-sky days, the relative humidity was in a narrow range of 26 % - 43 %, while in the cloud-topped days, this range was much wider 28 % - 77 %.

## 6    Conclusions

In this study, the dependency of the normalized vertical velocity variance, $\sigma_w^2/w_*^2$, on the meteorological conditions in the convective boundary layer is studied statistically during the summer periods (May-August) of the two consecutive years of the FESSTVaL 2020/21 field experiment. The mean day-to-day profiles were calculated from the raw Doppler lidar data at 1.5 s - 3 s resolution averaged over 30 min during the convective time of the day with a well developed boundary layer.

Similar to previous studies, we found that the mean $\sigma_w^2/w_*^2$ profile of all of the selected days is similar to the universal profile of Lenschow et al. (1980). However, daily mean profiles also show a high day-to-day scatter after normalization using the square of the convective velocity scale $(w_*^2)$. To investigate where this residual scatter originates from, we categorized the $\sigma_w^2/w_*^2$ profiles in two levels: first we separated clear-sky from cloud-topped and rainy days, and second, on clear-sky and cloud-topped days, we applied an additional level of categorization based on the relevant meteorological parameters.

The magnitude of the mean profile is highest during clear-sky days and systematically decreases in the cloud-topped and rainy days. We found that this change in the magnitude of $\sigma_w^2/w_*^2$ follows the changes in relative humidity in the boundary layer and the surface latent heat flux: $\sigma_w^2/w_*^2$ is lower in the case of a higher relative humidity and lower surface latent heat flux, and vice versa. This result suggests that the content of water vapor in the boundary layer and the vertical transport of moisture could explain most of the scatter in the observed profiles of $\sigma_w^2/w_*^2$. Since this dependency is a reversed one, the moisture content and the vertical transport of moisture are limiting factors on the intensity of turbulence in the convective boundary layer.

We further investigated the $\sigma_w^2/w_*^2$ profiles in two of the main categories separately, clear-sky and cloud-topped days. Since the effect of buoyancy is already taken into account in the scaling parameter $w_*$, we investigated other relevant meteorological parameters of the convective boundary layer.

In the clear-sky boundary layer, two regimes are found in the two different years, a low Bowen ratio regime with lower $\sigma_w^2/w_*^2$ and a high Bowen ratio regime with higher $\sigma_w^2/w_*^2$ magnitudes. These changes in $\sigma_w^2/w_*^2$ and its dependency on Bowen ratio are driven by different surface latent heat flux magnitudes in the two years. Besides Bowen ratio and surface latent heat flux, $\sigma_w^2/w_*^2$ profiles did not show any robust sensitivity to other parameters such as bulk stability or friction velocity under clear-sky conditions.





In the cloud-topped days, a systematic change in $\sigma_w^2/w_*^2$ profiles is found with the cloud fraction, surface latent heat flux, relative humidity and Bowen ratio. We found no dependency of $\sigma_w^2/w_*^2$ profiles on the bulk stability, while the highest $\sigma_w^2/w_*^2$ is found at the intermediate friction velocity ($u_*$) values. However, scaling of $\sigma_w^2$ by using a modified convective velocity that accounts for the effects of wind shear (Moeng and Sullivan, 1994) did not reduce significantly the variability of the daily profiles compared to the scaling using the square of the convective velocity scale ($w_*^2$).

Based on our results and comparison to the results of previous studies, we conclude that a systematic and robust dependency of $\sigma_w^2/w_*^2$ on cloud fraction across different locations and seasons could not be expected. This relationship will depend on the regime of soil moisture, because of a complex interplay of multiple competing mechanisms of the land-atmosphere interactions that lead to the formation of clouds. Therefore, cloud fraction is not an adequate parameter on its own for investigating the vertical profiles and magnitude of $\sigma_w^2/w_*^2$.

Our study confirms the conclusion made in Hogan et al. (2009) that shallow clouds are not a relevant source of turbulence in the convective boundary layer. Moreover, we find that the intensity of turbulence reduces with an increase in fraction of the boundary layer clouds, except in the cloud layer between approx. $0.9z_i$ and $1z_i$.

     The results of our study have an implication for the development of parameterizations of boundary layer turbulence and convection. The convective velocity scale, $w_*$, frequently used in these parameterizations does not account for important factors that control the intensity of turbulence in the convective boundary layer, expressed through the variance of vertical

velocity. To improve these parameterizations, a new scaling law has to be developed that will take into account the influence of moisture transport from the land surface to the atmosphere, content of water in the boundary layer and development of boundary layer clouds that is highly controlled by the state of the land surface.

*Data availability.* The Doppler lidars data used in this study are available at https://doi.org/10.25592/uhhfdm.10385 (Dewani and Leinweber,

2022). The atmospheric boundary layer measurement tower data are available from Deutscher Wetterdienst (DWD) upon request.

*Author contributions.* LS, MS, and ND participated in the planning of the FESSTVaL campaign. RL installed and performed the Doppler lidar measuerements. ND performed data processing. MS and ND designed the study, analyzed the result and prepared the manuscript. LS, JS, RL and MS provided input on the manuscript.

*Competing interests.* The authors declare that they have no conflict of interest.

*Acknowledgements.* We thank to Kevin Wolz (KIT, IMK-IFU), who provide the Level 1 of Doppler lidar data in Summer 2020, Ewan O'Connor (FMI) for providing a Doppler lidar in Summer 2021. We would thank to Frank Beyrich (DWD, MOL-RAO), Jan schween (University of Cologne, Institute for Geophysics and Meteorology), and Eileen Päschke (DWD, MOL-RAO) for helpful discussion on the





processing data. This research was funded by the Hans Ertel Centre for Weather Research of DWD (third phase, The Atmospheric Boundary Layer in Numerical Weather Prediction) Grant No. 4818DWDP4.



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
