# Peer review of "Dependency of vertical velocity variance on meteorological conditions in the convective boundary layer"

_Atmospheric Chemistry and Physics, 2022_

## Referee Comment (RC1)

**General Comments**

This manuscript examines daily profiles of normalized vertical velocity variance to explore where observed residual/unexplained variance originate. The authors focus on moisture effects and conclude that a new scaling parameter is needed for convective boundary layer parameterizations that account for such effects. The topic is relevant and the paper is generally well-written.

My first concern is that I believe the authors' formulation for $w_*$ is incorrect as presented in their Eq.(2). The Deardorff (1970) formulation is

$$w_* = \left[ \frac{g}{\theta_{vr}} z_i \left( \overline{w'\theta_v'} \right)_o \right]^{\frac{1}{3}}.$$

If we focus on the virtual heat flux (drop the subscript and assume surface values), we can expand as

$$\overline{w'\theta_v'} = \overline{w'\theta'} + 0.61\theta_{vr}\overline{w'q'}.$$

Using the authors' notation, we can define

$$\text{SHF} = \rho c_p \overline{w'\theta'} \qquad \text{and} \qquad \text{LHF} = \rho L_v \overline{w'q'}.$$

Substitution yields,

$$\overline{w'\theta_v'} = \frac{\text{SHF}}{\rho c_p} + 0.61\theta_{vr}\frac{\text{LHF}}{\rho L_v}$$
$$= \frac{1}{\rho c_p}\left( \text{SHF} + \frac{0.61\theta_{vr}c_p}{L_v}\text{LHF} \right).$$

If we take $\theta_{vr} = 300$ K, $c_p = 1004$ J kg$^{-1}$ K$^{-1}$, and $L_v = 2.5 \times 10^6$ J kg$^{-1}$,

$$\overline{w'\theta_v'} = \frac{1}{\rho c_p}\left( \text{SHF} + \frac{0.61(300 \text{ K})(1004 \text{ J kg}^{-1} \text{ K}^{-1})}{2.5 \times 10^6 \text{ J kg}^{-1}}\text{LHF} \right).$$

Solving and substitution yields:

$$w_* = \left[ \frac{g}{\theta_{vr}} \frac{z_i}{\rho c_p} \left( \text{SHF} + \mathbf{0.07}\text{LHF} \right) \right]^{\frac{1}{3}}.$$

Note that the coefficient is $0.07$, and not $0.7$ as listed in the manuscript's Eq.(2). The authors should check which version they used, because the use of $0.7$ would give latent heat flux an order-of-magnitude more importance to the normalizing value $w_*$, which would perhaps lead to the observed lack of profile collapse after scaling and may call into question the entire premise of the paper.

Next, I believe there are issues with the choice of considered fields and conclusions drawn from their use. The use of BR and RH are relative terms. Why did the authors also not look at more absolute terms such as mixing ratio or dew point temperature? I worry that the use of relative terms hide analysis of other important terms contained within. For instance, the authors note that in the cases where BR was lowest, the LHF was highest and so was the normalized variance. Just looking at Eq. (2), an increase in LHF would presumably lead to a larger $w_*$ in the absence of a known change in SHF. That in turn would lead to a lower value of the normalized variance values without a known change in the absolute variance, which is the opposite of the findings. In other words, it is hard to gauge any conclusions without knowing how related terms are affected in the presented scenarios. SHF is especially ignored throughout by the confusing justification that SHF is already accounted for in $w_*$. However, LHF is also contained in the equation, as shown above and in the authors' own Eq. (2). I think a more advanced multivariate analysis technique is needed to establish whether the stated conclusions are valid, especially in light of the potentially wrong form of $w_*$ as described in the paper.

Based on the above considerations, I believe this paper requires enough work that it would look substantially different than it does in its present form. Additionally, assuming that my mathematics are correct, this paper cannot be published nor conclusions trusted without knowing whether Eq. (2) contains a simple typo or the authors used an incorrect coefficient. In addition to these broad issues, I have a few specific issues that are listed below. Accordingly, I recommend that the manuscript be **rejected** for publication in *Atmospheric Chemistry and Physics*.

**Specific Comments**

Line 54  "*Large Eddy Simulations*" need not be capitalized here.

Eq. 2  See above for my comment, but I believe the expression for $w_*$ is wrong. There is also no citation or explanation of how the authors arrived at this expression.

Fig. 6  The caption should read "The outliers *are* denoted ..."

Line 150  In Figure 10, the authors present correlation coefficient with two digits to the right of the decimal. For Bowen ratio, the value is shown as $0.28$, yet the text says "*an absolute correlation coefficient equal to* $0.3$ ..." The authors should either present the correlation coefficients as rounded to the tenths spot in the figure, or write that the value is approximately equal to $0.3$ in the text.

Figure 10  Temperature has a similar absolute correlation coefficient as Bowen ratio. Why not examine that as well?

Line 162  The authors state that since "*the sensible heat flux is already taken into account in the convective velocity scale, the* $\sigma_w^2/w_*^2$ *dependency is now examined based on the surface latent heat flux classification.*" I am confused by this reasoning to avoid examining SHF because LHF is also accounted for in the convective velocity scale (per Eq. 2).

Sect. 4.2.3  Given that there is no analysis for rainy days, this short section seems unnecessary. The authors even allude to this on Line 110.

Line 195  Again, I am confused why the scaling by $w_*$ means that the dependency of the normalized variance on the Bowen ratio is attributable only to LHF. Both SHF and LHF are contained in $w_*$.

---

## Author Comment (AC1)

**Response to Referee Comments 1 (RC1) on ACP-2022-543 "Dependency of vertical velocity variance on meteorological conditions in the convective boundary layer" - by Noviana Dewani et al.**

We thank the referee for all the comments and suggestions. The comments are copied below and our point-to-point responses to the comments are presented in blue. The main three points in our responses are summarized as follows: First, the typo in equation (2) is now corrected in the manuscript. We also provide the script that we used for the analysis to demonstrate that the analysis was not affected by this error in writing. Second, we demonstrate that the contribution of LHF in the calculation of $w_*$ can be considered negligible compared to the contribution of SHF. We also show why SHF could not be used to explain the remaining variability in the $w'^2/w_*^2$. Third, we present a more detailed analysis using multiple variables and their co-dependencies as shown in the correlogram (Fig. 14) in the revised manuscript. We have also updated Fig. 9 in the revised manuscript to justify our choice of the parameters that we used for the first and second levels of categorization based on the meteorological conditions. We hope that our answers, clarifications and additional analysis would suffice to qualify our manuscript for further consideration for publication in ACP.

**1 General Comments**

This manuscript examines daily profiles of normalized vertical velocity variance to explore where observed residual/unexplained variance originate. The authors focus on moisture effects and conclude that a new scaling parameter is needed for convective boundary layer parameterizations that account for such effects. The topic is relevant and the paper is generally well-written.

My first concern is that I believe the authors' formulation for $w_\star$ is incorrect as presented in their Eq.(2). The Deardorff (1970) formulation is

$$w_* = \left[ \frac{g}{\theta_{vr}} z_i \overline{(w'\theta')_o} \right]^{\frac{1}{3}}$$

If we focus on the virtual heat ux (drop the subscript and assume surface values), we can expand as

$$\overline{w'\theta'_v} = \overline{w'\theta'} + 0.61\theta_{vr}\overline{w'q'}$$

Using the authors' notation, we can define

$$\text{SHF} = \rho c_p \overline{w'\theta'} \quad \text{and} \quad \text{LHF} = \rho L_v \overline{w'q'}$$

Substitution yields,

$$\overline{(w'\theta'_v)} = \frac{\text{SHF}}{\rho c_p} + 0.161\theta_{vr}\frac{\text{LHF}}{\rho L_v}$$

$$= \frac{1}{\rho c_p}\left( \text{SHF} + \frac{0.61\theta_{vr}c_p}{L_v}\text{LHF} \right)$$

If we take $\theta_{vr} = 300\,K$, $c_p = 1004 J\,Kg^{-1}\,K^{-1}$, and $L_v = 2.5x10^6\,J\,kg^{-1}$

$$\overline{(w'\theta_v')} = \frac{1}{\rho c_p}\left(\text{SHF} + \frac{0.61(300\,K)(1004 J\,Kg^{-1}\,K^{-1})}{2.5x10^6\,J\,kg^{-1}}\text{LHF}\right)$$

Solving and substitution yields:

$$w_* = \left[\frac{g}{\theta_{vr}}\frac{z_i}{\rho c_p}(\text{SHF} + \mathbf{0.07}\text{LHF})\right]^{\frac{1}{3}}$$

Note that the coefficient is 0.07, and not 0.7 as listed in the manuscript's Eq.(2). The authors should check which version they used, because the use of 0.7 would give latent heat flux an order-of-magnitude more importance to the normalizing value $w_\star$, which would perhaps lead to the observed lack of profile collapse after scaling and may call into question the entire premise of the paper.

We admit that there was a typo in the manuscript. The coefficient was supposed to be written as 0.07 instead of 0.7 as written in Eq.2. The equation for the surface fluxes is derived from equation 2.80 in Garratt (1994) which is defined by

$$\overline{(w'\theta_v')_0} = \overline{(w'\theta')_0}\left(1 + 0.61\bar{\theta}\gamma/B\right)$$

with $B = c_p\,\overline{(w'\,\theta')_0}/\lambda\overline{(w'\,q')_0}$,

When B is substituted in equation 2.80, we get

$$\overline{(w'\,\theta_v{'})_0} = \overline{(w'\,\theta')_0} + 0.61\,\bar{\theta}\,\overline{(w'\,q')_0}$$

and with the relation between kinematic and dynamic fluxes written as

$$\overline{(w'\,\theta')_0} = SHF/c_p\rho$$

and

$$\overline{(w'\,q')_0} = LHF/L_v\rho$$

,

with $c_p = 1005\,J\,K^{-1}\,K$, $\quad\bar{\theta} = 288.15\,K\quad and\,L_v = 2.45\,10^6\,J\,K^{-1}$, the surface fluxes become

$$\overline{(w'\,\theta_v')_0} = \frac{1}{c_p\rho}\left(SHF + 0.07\,LHF\right)$$

The python code which is used for this study is available in dewani (2022),

```
buoy = (hfss+(0.07*hfls))/(rho*cp)
w_star = ((9.81/np.array(Tv30min))*buoy*z1)**(1/3)
```

Next, I believe there are issues with the choice of considered fields and conclusions drawn from their use. The use of BR and RH are relative terms. Why did the authors also not look at more absolute terms such as mixing ratio or dew point temperature? I worry that the use of relative terms hide analysis of other important terms contained within.

We have looked into absolute humidity and additional terms and provide a more complete analysis in Fig. 1 below. For the first level of categorization, we use typical conditions observed during summer at the measurement site, predominantly clear sky days, days with the presence of clouds but with no rain events, and the days with rain in a separate category. We find a good correlation (absolute coefficient correlation = 0.4 or above) between $w'^2/w_*^2$ and moisture variables: absolute humidity, LHF, cloud fraction and RH. This is the reason why we proceed to analyse the vertical profiles of $w'^2/w_*^2$ using these variables as criteria for our further levels of classification. We further demonstrate how the

normalized variance profile changes with the absolute humidity using three categories based on the absolute humidity (Fig. 8 in the revised manuscript showing all cases and Fig. 2 below showing the results for the cloudy days only). Fig. 8 in the revised manuscript shows the variance profiles with the highest normalized variance magnitude in the clear sky days (lowest absolute humidity). The average profiles of the absolute humidity in the cloudy and rainy category are showing similar magnitudes, while in the plots showing relative humidity cloudy and rainy categories are distinct. The lower magnitude of the normalized variance in the rainy days shows how the rain and associated processes affect the turbulence in the ABL. The relative humidity (RH) values reflect these effects of the rain and most likely also downdraft and cold pool effects in the ABL as the three categories are clearly distinguishable in the RH plot (Fig. 8, middle, in the revised manuscript). When the cloudy days with no rainy events are analysed separately (Fig. 2), we find that the magnitude of the normalized variance is the lowest in the category with a high absolute humidity compared to the categories with a low and medium absolute humidity. Although, the profiles of the lower and medium absolute humidity categories are similar in the lower elevation.

[Figure]

Figure 1: Scatter plot of the magnitude of normalized variance and meteorological parameters. Blue dots represent the data in the clear-sky days, orange dots represent the data in cloud-topped days and red indicate the data in the rainy days.

For instance, the authors note that in the cases where BR was lowest, the LHF was highest and so was the normalized variance. Just looking at Eq. (2), an increase in LHF would presumably lead to a larger $w_\star$ in the absence of a known change in SHF. That in turn would lead to a lower value of the normalized variance values without a known change in the absolute variance, which is the opposite of the findings.

We disagree on this point. To explain our understanding of this matter, we would like to emphasize two results here:

1) Mathematically speaking from Eq. (2), the contribution of the latent heat flux to $w_*$ can be considered negligible. In the surface heat flux equation, SHF is taking more role than LHF which is multiplied by a factor of 0.07. It means that the convective velocity mostly depends on SHF with small contribution from LHF. Our plots below clearly show that the convective velocity scale mostly depends on SHF and the contribution of LHF is negligible (Fig. 3).

[Figure]

Figure 2: Dependency of the normalized vertical velocity variance profiles on the absolute humidity and Bowen ratio. The average absolute humidity, BR, LHF, SHF and $w_*$ are shown as reference for the related profiles.

2) We demonstrate in Fig. 4 that after the normalization, the variance profiles stay very similar in the three categories based on the SHF, while they strongly depend on the LHF. Therefore, our conclusion stands that the dependency of the normalized variance on the Bowen ratio reflects its dependency on the LHF rather than SHF.

In other words, it is hard to gauge any conclusions without knowing how related terms are affected in the presented scenarios. SHF is especially ignored throughout by the confusing justification that SHF is already accounted for in $w_\star$. However, LHF is also contained in the equation, as shown above and in the authors' own Eq. (2).

A small contribution of LHF to $w_*$ can be considered negligible. We show this in Fig. 3, as a good correlation is found between $w_*$ and SHF while a poor correlation exists between $w_*$ and LHF. We also show in Fig. 4a that the classification based on SHF results in the normalized variance profiles that are very close to each other. Based on this, we conclude that after the normalization, SHF no longer influences the normalized variance profiles as the normalized profiles are similar in respect to SHF. This happens because SHF is the main contributor to the near-surface buoyancy flux used to calculate $w_*$.

[Figure]

Figure 3: (Left) Comparison between convective velocity scale and sensible heat flux; (right) convective velocity scale compared to the latent heat flux.

I think a more advanced multivariate analysis technique is needed to establish whether the stated conclusions are valid, especially in light of the potentially wrong form of $w_\star$ as described in the paper. Although we cannot perform a comprehensive statistical multivariate analysis given the limited data set at hand, we have however examined joint distributions and co-dependencies between variables at hand. To demonstrate the co-dependencies between the main factors considered here, we show correlograms in Fig. 14 in the revised manuscript. Besides showing the co-dependencies between the main meteorological variables, the correlograms also show the joint distributions of these variables and the normalized variance shown as colored scatter points. The analysis of the correlogram also point to the moisture related variables as controlling factors of the normalized vertical velocity variance. As we already mentioned, there was a typo in the $w_*$ equation in the submitted version of the manuscript.

Based on the above considerations, I believe this paper requires enough work that it would look substantially different than it does in its present form. Additionally, assuming that my mathematics are correct, this paper cannot be published nor conclusions trusted without knowing whether Eq. (2) contains a simple typo or the authors used an incorrect coefficient. In addition to these broad issues, I have a few specific issues that are listed below. Accordingly, I recommend that the manuscript be rejected for publication in Atmospheric Chemistry and Physics.

We present the revised manuscript where we took into account all the comments of the reviewer. The revised manuscript includes a more detailed analysis and plots that we initially did not include, such as plots showing the sensible heat flux and absolute humidity. The results and conclusions of our study did not change significantly based on this extended analysis and considerations. We have also provided our original code along with the data used in the analysis showing that the typing error was only present in the manuscript text and did not affect our results. We invite the reviewer to inspect our code and data used for the analysis that are available at the provided links. We hope that our

[Figure]

Figure 4: Profiles of the mean normalized velocity variance classified by (a) sensible heat flux and (b) latent heat flux. The corresponding convective velocity scale, sensible heat flux and latent heat flux are shown in barplots.

manuscript in its present revised form can be considered for publication.

**2   Specific Comments**

Line 54 : "Large Eddy Simulations" need not be capitalized here. We thank the reviewer for this suggestion and have corrected the text accordingly.

Eq. 2 : See above for my comment, but I believe the expression for $w_*$ is wrong. There is also no citation or explanation of how the authors arrived at this expression. The $w_*$ equation in the manuscript has a typo. We used the surface heat flux equation as shown in the revised manuscript as below

$$(\overline{w'\,\theta_v'})_0 = \frac{1}{c_p\rho}\left(SHF + 0.07\,LHF\right)$$

.

Fig. 6 : The caption should read "The outliers are denoted ..."
We thank the reviewer and have made the corrections.

Line 150: Figure 10, the authors present correlation coefficient with two digits to the right of the decimal. For Bowen ratio, the value is shown as 0.28, yet the text says "an absolute correlation coefficient equal to 0.3 ..." The authors should either present the correlation coefficients as rounded to the tenths spot in the figure, or write that the value is approximately equal to 0.3 in the text
We replaced the old Fig. 10 as well as its explanation in the text with two new figures, Fig. 9 showing the scatter plots between the normalized variance and various meteorological parameters (we have extended the list of the analysed parameters) and Fig. 14 in the revised manuscript, which shows co-dependencies and bivariate analysis in cloud-topped days.

Figure 10 : Temperature has a similar absolute correlation coefficient as Bowen ratio. Why not examine that as well?
Temperature has a relatively low absolute correlation coefficient (about 0.28) to the magnitude of the normalized variance. Previously, we investigated further the variables that had an absolute correlation

coefficient higher than 0.3. The dependency on the BR in cloud-topped days was investigated in addition because we wanted to see whether we can also find a similar dependency between the normalized variance and BR like in the clear-sky days. However, we do not emphasize the importance of BR as it likely just reflects the dependency of the normalized variance on LHF. Nevertheless, we changed Fig. 10 and refer to Fig. 9 in the revised manuscript for the selection of the parameters used in the further analysis.

Line 162 : The authors state that since "the sensible heat flux is already taken into account in the convective velocity scale, the $\sigma_w^2/w_*^2$ dependency is now examined based on the surface latent heat flux classification." I am confused by this reasoning to avoid examining SHF because LHF is also accounted for in the convective velocity scale (per Eq. 2).
We admit that this was an inadequate wording. What we intended to point out here is: the LHF contribution in the surface heat flux equation for $w_*$ is very small as we show in Fig. 3 above. It means that SHF has the main contribution in the calculation of $w_*$. We have also observed that when we use the classification of the cases based on the SHF values, the vertical profiles of the mean normalized variance are very similar among the categories, which means that the SHF can not explain the residual variability of the normalized variance profiles (Fig. 3). We rewrite the sentences in the revised manuscript line 178: "Since the contribution of latent heat flux to $w_*$ is negligible compared to the contribution of sensible heat flux, we examined if there is any remaining dependency of the normalized profiles to latent heat flux, as the Fig. 9 suggests."

Sect. 4.2.3 : Given that there is no analysis for rainy days, this short section seems unnecessary. The authors even allude to this on Line 110.
We would like to show how the day-to-day variability looks like in the rainy days because we used the mean profile of rainy days in comparison with clear-sky days and cloud-topped days as written in Line 110. However, we did not perform a detailed analysis for rainy days because more factors might be playing a role in the variability of the variance profiles that would be out of scope of our current study. We exclude the rainy days section in our revised manuscript.

Line 195: Again, I am confused why the scaling by w means that the dependency of the normalized variance on the Bowen ratio is attributable only to LHF. Both SHF and LHF are contained in $w_\star$.
The SHF is the main contribution in $w_*$, and the $\sigma_w^2/w_*^2$ profiles are similar in respect to the SHF classification. It implies that the influence of SHF on the $\sigma_w^2/w_*^2$, is eliminated by the normalization using the convective velocity scale. We can also see in the Bowen ratio classification (Fig. 2), that the sensible heat flux (Fig. 2f) has a similar pattern to convective velocity scale (Fig. 2g), meaning that the remaining variability in the profiles in respect to the Bowen ratio classification can be attributed to the variation in LHF.

**References**

dewani. dwni/vert_mode:, Aug. 2022. URL https://doi.org/10.5281/zenodo.6947702.

J. Garratt. *The Atmospheric Boundary Layer*. Cambridge Atmospheric and Space Science Series. Cambridge University Press, 1994. ISBN 9780521467452.

**Response to Referee Comments 2 (RC2) on ACP-2022-543 "Dependency of vertical velocity variance on meteorological conditions in the convective boundary layer" - by Noviana Dewani et al.**

We thank the referee for their insightful and constructive comments and suggestions. The comments are copied below and our point-to-point responses to the comments are presented in blue.

In summary of our responses, we list here the three major points: 1) The estimation of the mixing layer height (MLH) is revised and the comparison between the retrievals from the two Doppler lidars used in 2020 and 2021 is presented in detail. According to the comparison of the MLH and $w_*$ between the two different instruments as suggested by the referee, the two lidars now show good agreement. So, we conclude that the two data sets from 2020 and 2021 can be analysed jointly; 2) Relative versus absolute variables as classification criteria are revised and explained. We have added absolute humidity into the analysis and also show the plots of several additional variables in the revised manuscript. Additional scatter plots are added based on which we selected the main classification criteria; and 3) The relationship between the vertical velocity variance and soil moisture is further investigated.

The changes made to the manuscript are also attached beside this response.

**1 Referee Comments 2:**

The authors have presented a clear, well-written discussion of the characteristics of scaled vertical velocity profiles for different environmental conditions as measured by Doppler lidars at the Lindenberg site. However, their analysis depends on some assumptions that have not been fully tested and their classification method possibly obscures scientifically useful results. Because of these issues, I believe that the paper needs major revisions before acceptance by ACP.

**1.1 Major revisions**

The first issue pertains to the use of two different instruments for the analysis. The authors note that there is substantial agreement between the two Doppler lidars when they were co-deployed for three days, as quantified by the strong correlation of the observed vertical velocity at a single range gate. However, observations of the individual vertical velocities are less important than the derived values of w* and ML depth. With the different instrument subject to different SNR filtering, different ML calculations, different near-surface cutoffs, and different w* calculations, correlation between the derived quantities is not guaranteed. The authors' implicit claim that the observations between these two systems can be interchanged would be stronger if a comparison were done between the actual observations being used for the bulk of the analysis. While it is likely that there is a causal relationship between the different Bowen ratios and the different profile behaviors in the two analysis years, the fact that different instruments with different processing pipelines were used in each of the two years cannot be overlooked as a potential source of the observed changes. A new Fig. 3 that showed w* and ML values instead of w values at a single height would help validate the claim that the two systems produce interchangeable observations.

We revised the ML height algorithm in order to improve the estimated ML height based on the retrievals from the two instruments employed in concurrent time. We still used the variance method to estimate the ML height, however the threshold is changed from 0.04 to 0.09 for DL161, which is the same threshold already applied on the DL146 dataset. So, now both datasets use the same threshold for the estimation of the ML height. Figure 1 shows a comparison of $w_*$ and ML height between the two devices during the concurrent dates and only within the analysis time period from 10 UTC to 15 UTC. Due to a rain event on 25 July 2022, we only use the time from 10UTC - 14.30 UTC during this day. A good correlation is shown between the ML height and $w_*$ obtained from the two devices. This indicates that the two devices can provide interchangeable observations. If we look at the daily profiles of the normalized variance using the normalized height at the y axis, as shown in Fig.2a, the profiles derived from the measurements of the two devices fall close to each other. With the old threshold, the normalized variance has slightly different values due to different ML heights (Fig.2b). Now, with the same threshold used to estimate the ML height, this difference is significantly reduced. The normalized variance profiles match very closely between the two instruments on 24 and 25 July 2021. We replaced the analysis using the Doppler lidar in Fig. 3 with this new one and the subsequent analysis is redone in the revised manuscript (lines 88-93).

[Figure]

Figure 1: Comparison between the mixing layer height and $w_*$ derived from two co-located Doppler lidars during three days, the 23., 24. and 25. July 2021 using 0.04 and 0.09 as a threshold for the estimation of the MLH. The scatter plots represent the comparison between DL161 and DL146 using 0.09 as a threshold.

[Figure]

Figure 2: Comparison between co-located DL161 and DL146. (a) Daily profiles of vertical velocity variance without normalization; Daily profile of normalized vertical velocity variance with different threshold value for estimate MLH on DL161 (dashed-line): (b) 0.04 and (c) 0.09. The profiles are averaged between 10UTC-15UTC for 23 and 24 July 2021 while between 10UTC - 14.30UTC for 25 July 2022.

The second significant issue is the classification of analysis days into clear, cloud-topped, and rainy. This sorting can obscure the actual processes at work: rain can come from convectively-driven clouds forced by PBL processes, or it can come from stratiform clouds with no real connection to the PBL; ...

We choose the three main categories as the most typical weather conditions that can be found during summer: clear sky or cloudy (regardless of the cloud type but without rain during the day) and the days with mostly convective rain events as an additional category, but not in focus for this study. We keep the rainy category because we find it interesting and relevant to show how the convective rain events change the shape of the variance profile in the ABL. Most importantly, the separation of the cloudy days into the days with precipitation and the days without precipitation served to exclude the impact of the rain on the ABL turbulent processes. We emphasize in the revised manuscript (line : 113) : "The impact of rainy events on the ABL turbulence is not the focus of our study and would require a separate investigation."

...convective boundary layers might or might not form clouds depending on conditions both within and outside of the PBL. One reason some of the results show little to no dependence of the scaled w variance profiles on different environment types are occurring could be because different processes are being aggregated into the same bins while the same processes are being distributed into different ones.

We do not base our conclusions solely on the main three categories but we take a step further and apply a second level of classification to clear-sky and cloudy days using most of the available measurements at the site. Moreover, we did further analysis on the correlation between the normalized variance and the main meteorological parameters to investigate the impact of surface conditions, conditions within and above the ABL, as well as cloud fraction on the variability of normalized variance profile as shown in Fig. 5. To investigate the influence of the conditions above the ABL, we examine potential temperature and mixing ratio from radiosonde measurements at 12 UTC to derive the values at the mixing layer height level, a free-tropospheric level , free-tropospheric lapse rate as well as $\Delta\Theta$ and $\Delta qv$ as shown in Fig. 3. The relationship between the normalized variance with potential temperature and mixing ratio shows no significant correlation. The normalized variance is mostly influenced by the ABL and surface conditions. We also looked at co-dependencies and joint distributions between the main factors at hand (Fig. 14 in the revised manuscript). Both of these scatter plots and correlograms show that the normalized variance shows good correlation with moisture variables. We base the choice of the parameters for further analysis based on the scatter plots and the correlogram.

The authors should go into some detail about why these specific categories were chosen over more robust measures of PBL turbulence that could be quantified using other instruments at the Lindenberg site.

Our focus is on the PBL turbulence characterized by the vertical velocity variance on the clear-sky days and cloud-topped days and how it relates to environmental conditions. For this analysis, we use a categorization based on the main meteorological parameters to characterize turbulence, instead of using other turbulence measurements conducted during FESSTVaL to make a categorization. Other ABL turbulence-related projects are already ongoing within the FESSTVaL community and the results of these studies could be used in a subsequent study where different measures of turbulence would be inter-compared and our study could be extended. This however, is not the current topic of your work.

**1.2 Minor issues**

Line 77: How, specifically, were clouds identified? Is it a backscatter threshold from the Doppler lidar? What role did the ceilometer have in this?
We used attenuated backscatter with a specific threshold for cloud identification as mentioned in the submitted manuscript line 104. To confirm the clouds identified using the Doppler lidar, we do a comparison with the cloud base identified using the ceilometer, as mentioned in the submitted manuscript in line 77.

Line 80: Given that the two instruments were both Halo Streamline XR Doppler lidars, why were they treated differently in processing, with different noise filtering and other changes?

[Figure]

Figure 3: Correlation between normalized vertical velocity variance and temperature potential as well as mixing ratio. (a) potential temperature at mixing layer height; (b) free-tropospheric lapse rate of potential temperature; (c) potential temperature at free-tropospheric level; (d) potential temperature jump. Figure (e), (f), (g) and (h) are similar variable but for mixing ratio.

To determine the SNR filter threshold for both lidar datasets, first we looked at the plot of the intensity against vertical velocity as shown in Fig. 4. The plots show a different intensity (SNR+1) band for the distributed vertical velocity. Therefore, we could not use the same SNR threshold for both Doppler lidars. In addition, we applied the additional SNR filtering to DL161 because we would like to obtain as much data as possible in the upper elevation which we could not obtain if we only used the SNR threshold. Several first range gates are also removed because of the noise that we can see as a straight line on 15 m/s of the vertical velocity retrieved from the DL146 (Fig. 4). We included the figure into the revised manuscript as Fig. 2.

Line 98: Related, why different ML calculations?
We re-investigated the MLH threshold for both datasets. Previously, the estimated MLH using 2020 dataset threshold (0.04) for the 2021 dataset produces quite inconsistent and unreasonable results on some dates in the 2021 dataset. We do re-analysis to find the best threshold by apply some different thresholds to both datasets and we decided to use 0.09 as a threhold for both dataset. The comparison

[Figure]

Figure 4: Vertical velocity vs Intensity for DL161 (left) and DL146 (right) without any data filter applied.

of the results between the two datasets now show a good agreement as shown in Fig. 1b. We re-processed all datasets using 0.09 threshold for the analysis. The results on the dependency of the variance profiles on meteorological conditions after the re-calculation are not significantly different. So, the results of the current analysis still align with our conclusions.

Line 105: What is a "day" for the purpose of this analysis? Is it a 24 h period? Sunrise to sunset (meaning it's different for each day in the analysis)? A fixed period of time as shown in Figs. 4 and 5? We use the same time period on each day for the analysis which is from 10UTC to 15UTC. This is the time period of a well developed boundary layer, excluding the morning and evening transitions. All of the variables are calculated within this time window. We emphasize in the line 116 in the revised manuscript: "The days with a cloud fraction above 0.003 within time analysis period were categorized as cloud-topped."

Line 107: Can you explain the value of finding a characteristic scaled variance profile for cloudy days regardless of cloud type when certain cloud types are going to be closely coupled to PBL processes? We investigated the variability of the normalized variance profile in cloudy days during summer over land, which are mostly convective-driven. We choose to analyse the cloudy days regardless of the cloud type in order to compare the normalized variance profiles to the universal profile of Lenschow et al. (1980). Most of the previous studies used the universal profile of Lenschow et al. (1980) as a reference profile, not only for comparison using observational data but also using model results. In these previous studies as well as in the original paper of Lenschow et al. (1980), the measurements took place during various conditions, not only in the convective ABLs, but also during cold air outbreaks with scattered to broken stratocumulus in the upper part of the mixed layer. The purpose of our investigation is to characterize the variability of our measurements of the vertical velocity variance, how the average profile during summer at the Falkenberg site compares to the reference profile regardless of cloud type and how the variability in the profiles relates to the main meteorological parameters. Knowing the variability and its dependency on the environmental conditions could help to make an improvement of empirical scaling.

Line 131/Fig. 8: Seeing as there are rather large overlaps between the latent heat fluxes in the three categories (implying that absolute moisture fluxes have little to no impact), why is there such a substantial difference by relative humidity? It seems that rather than relative humidity having an impact on the mean profile, the relative humidity is more an indicator that an environment is clear/cloudy/rainy. Is there a causal relationship between RH and the scaled profile? Is RH actually the best measure when these processes would seeminly depend on absolute moisture quantities? We agree with the referee that the RH is an indicator of the clear sky, cloudy or rainy conditions and that these conditions affect the normalized variance profiles. We expanded our analysis to look further into the absolute humidity and other ABL conditions. We show the absolute humidity corresponding to the three main categories in Fig. 5d and Fig. 5f. The absolute humidity shows a similar trend as relative humidity here with a very large range of values in cloud-topped days. We also found that the absolute humidity and normalized variance have a good agreement (Fig. 6) besides relative humidity, cloud fraction and latent heat flux. Although there is a clear distinction between clear-sky and cloudy

days in the absolute humidity profiles, the distinction between cloudy and rainy days is not so obvious. This implies that the RH profiles clearly show the impact of rain, as the referee suggested.

[Figure]

Figure 5: Average profiles of $\sigma_w^2/w_*^2$ in the three main categories: clear-sky, cloud-topped and rainy days. The universal profile is given as the black dashed line. Box plots of the (b) surface latent heat flux (LHF), (c) relative humidity and (d) absolute humidity (RH) averaged over the days in the three categories; vertical profiles of (d) relative humidity and (e) absolute humidity at 12 UTC in the boundary layer averaged over the three main categories from radiosonde data (https://weather.uwyo.edu/upperair/sounding.html).

[Figure]

Figure 6: Scatter plot of the magnitude of normalized variance and meteorological parameters. Blue dots represent the data in the clear-sky days, orange dots represent the data in cloud-topped days and red indicate the data in the rainy days.

Lines 169-170: Can you explain why this is happening?

We found that the low surface latent heat flux is a main cause for the high Bowen ratio instead of very high sensible heat flux which probably corresponds to the soil moisture condition. We might be observing a transition between an energy-limited to water-limited regime, where the BR can be increased only in the case when the evapotranspiration is limited by low soil-moisture availability, as we most likely did not observe high-enough surface radiation to sufficiently increase the sensible heat flux to lead to such high BR values. Although this observation is very interesting, it would require further investigation that would distract from the focus of our study.

Line 221: This sentence seems to downplay the importance of the cloud-fraction analysis performed in this paper. Since soil moisture is important to these processes, it seems like a robust analysis of the scaled profiles would include soil moisture characteristics. Are there soil moisture observations at Falkenberg that could be used to illustrate the dependency of the profiles on soil moisture?
We demonstrate a comparison using soil moisture at the 8cm of depth at the Falkenberg site. The normalized variance values have a low dependency on the soil moisture as shown in Fig. 7. However, the soil moisture seems to have an indirect impact through the moisture transport such as the surface heat flux.

Furthermore, there is a crucial distinction in the analysis to be considered here: 1) the dependency between the normalized variance and cloud fraction at a single location in Lindenberg under an invariant soil moisture regime and 2) the dependency between the normalized variance and cloud fraction at the two different sites in different soil moisture and climate regimes. The point we would like to make in this paper is that although we find a good correlation between the normalized variance and cloud fraction at Falkenberg, this finding cannot be generalized to other regimes and locations. So, the cloud fraction cannot be used on its own in the analysis of the vertical velocity variance as is usually done in many studies. Although this explanation might be downplaying our analysis based on cloud fraction, it also explains why we observe the opposite change in the normalized variance with an increase in cloud fraction compared to the previous studies. We still include the cloud fraction analysis because it has been frequently used in previous studies.

Line 223: SGP, not GPS
We thank to the reviewer for the correction.

[Figure]

Figure 7: Scatter plot showing the dependency of the meteorological parameters on soil moisture in cloud-topped days. The dot colors indicate the values of the normalized vertical velocity variance.

---

## Author Response (AR2)

**Response to Referee Comments 2 (RC2) on ACP-2022-543 "Dependency of vertical velocity variance on meteorological conditions in the convective boundary layer" - by Noviana Dewani et al.**

**1 Referee Comments 2:**

My concerns are largely assuaged by the changes the authors made. I appreciate the effort they made in quantifying the similarity between the two Doppler lidar systems, the new emphasis on absolute measures of humidity instead of relying on only relative ones, and the investigation into soil moisture. I feel that the revised paper meets my expectations for publication, and have only notices a few small items for correction.

We thank the referee for taking the time to give some input and feedback to help us improve the quality of the manuscript. The changes made to the attached manuscript are according to the comments below.

Line 22: height, not heigh
We thank the reviewer to point this out. We made the correction on the revised manuscript.

Line 103: acceleration, not accelaration
We thank the reviewer to point this out. We made the correction on the revised manuscript.

Fig 14: Can you put r values in the corner of the panels? I can see that RH appears to have stronger correlations than other parameters, but it would be good to know how much. Right now the text merely says "good correlation" which can be very subjective.
We put the coefficient correlation between two parameters on the Fig. 14 and add in text line 216 : "and the coefficient correlation indicates the correlation between two parameters.".

Line 250: Comma not needed after although.
We thank the reviewer to point this out. We made the correction on the revised manuscript.